# Modifiable and nonmodifiable factors associated with anxiety, depression, and stress after one year of the COVID-19 pandemic

**Azharul Islam**[1¤]*, **Papia Mahbuba**[1], **Tanvir Ahmed**[1], **Shamsul Haque**[2]

**1** Department of Educational and Counselling Psychology, University of Dhaka, Dhaka, Bangladesh,
**2** Department of Psychology, Jeffrey Cheah School of Medicine and Health Sciences, Monash University Malaysia, Selangor Darul Ehsan, Malaysia

¤ Current address: Department of Educational and Counselling Psychology, Faculty of Biological Sciences, University of Dhaka, Dhaka, Bangladesh

* azharulislam@du.ac.bd

**Data Availability Statement:** The datasets generated during and/or analysed during the current study are available in a public repository. Weblink: https://doi.org/10.17632/6szgbkpzn8.1.

## Abstract

### Background

People worldwide have experienced various mental health issues during the COVID-19 pandemic. This study investigates the modifiable and nonmodifiable predictors of anxiety, depression, and stress among Bangladeshi participants after one year of the pandemic.

### Method

A large group of adult participants ($N$ = 1897), recruited from eight administrative divisions in Bangladesh, completed an online survey in May and June 2021 when the Movement Control Order was in place. We used the Beck Anxiety Inventory, Patient Health Questionnaire-9, and Perceived Stress Scale-4 to assess the participants' anxiety, depression, and stress. We also gave the Mindful Attention Awareness Scale and Life-Orientation Test-Revised to assess mindfulness and optimism.

### Results

The results revealed that the prevalence rates for anxiety and depression were 62.5% and 45.3%, respectively. Multivariate analyses showed that several nonmodifiable factors, such as those who were students, unmarried and females, and those living in the Northern region (Rajshahi and Mymensingh division) and dwelling in the rural areas, suffered from worse mental health (accounted for 5%-23% of the variances in the mental health outcome scores). Modifiable factors accounted for an additional 10%-25% of the variances in the same outcome variables. Adults with higher mindfulness and optimism, living in the country's Southern region (Chattogram division) and those who took both vaccine doses and had no history of mental illness reported better mental health.

**Funding:** The research data collection has been supported by Dhaka University's Centennial Research Grant (Reg/Admin-3/70924) awarded to Dr. Azharul Islam. The funders had no role in the study design, data collection, analysis, the decision to publish, or the preparation of the manuscript.

**Competing interests:** The authors have declared that no competing interests exist.

## Conclusion

Anxiety, depression, and stress remained high in Bangladeshi adults after one year of the pandemic. The community-based interventions should aim to increase the mindfulness and optimism levels among the sufferers. More accelerated vaccination programs across the country could protect people from suffering from overall mental distress.

## Introduction

The coronavirus disease-19 (COVID-19) pandemic seems to persist for a longer period. There is no forecast on when the pandemic might end, and life will return to normal as a new variant of the virus emerges regularly. The existential threat posed by the virus, sudden changes in daily routine and mass quarantine, worry about close relatives being infected, and the economic uncertainty resulting from the total or partial closure of business activities have significantly affected the mental health of the world population [1, 2]. Data from countries indicate that people increasingly report mental health concerns, particularly elevated anxiety, depression, and stress. The global prevalence of stress among the general adult population in the first seven months of the pandemic ranged from 29.6% to 38%, anxiety from 26.9% to 50.9%, and depression from 28.0% to 48.3% [3–5]. The immediate psychological reactions to the pandemic across the countries were consistent and worse than the pre-pandemic situation [1, 5].

Many studies on different world populations have reported that people suffered from various mental health problems during previous epidemics (see Zurcher et al. for a comprehensive review [6]). For example, Kamara and colleagues [7] found 12% of their participants recruited from Sierra Leone to be suffering from moderate to severe emotional disorder or depression during the West African Ebolavirus epidemic in 2014–2016. Xu and colleagues [8] studied 1082 Chinese college students and found that 2% suffered from symptomatic PTSD during the 2009 H1N1 influenza pandemic. Quah and Hin-Peng [9] studied crisis prevention and management during the 2002–2004 SARS outbreak in Singapore and reported that 2.9% of their respondents suffered from high and 42.4% from moderate anxiety levels.

An unprecedented health emergency with a high fatality rate during the COVID-19 pandemic has panicked many people [10]. At the outset of the pandemic, there was no vaccine, and the most appropriate interventions available were enforcing various degrees of movement restrictions, maintaining social distance, wearing a facemask, and frequent handwashing. The lack of adequate and authentic information and unprepared health systems in many countries caused significant concern and worry among citizens and healthcare workers [9].

However, research has shown that mental health reactions to this challenging health emergency were not equally detrimental to all; some still maintained a healthy state. For example, Pierce, McManus [11] tracked the mental health outcomes of 19,763 UK adults from April to October 2020. At the onset of the pandemic, the average mental health of the British people declined, but it started bouncing back in July 2020. The authors found that people had either consistently good (39.3%) or consistently very good (37.5%) mental health over the first six months of the pandemic. Individuals with a history of physical or psychological illness, those who lived in deprived neighbourhoods, or members of Asian, Black, or mixed-ethnic communities were more likely to report worse mental health as the pandemic progressed [1].

As the COVID-19 pandemic and the resulting threats and uncertainties differentially affected citizens' mental health, it is crucial to identify modifiable factors that could protect people's mental health during this pandemic. Nonmodifiable personal attributes, such as age, sex, education, and marital status, are also worth investigating. Modifiable factors are those

factors that we could address (thus modified) through psychosocial interventions. Two personality traits likely to be modified, such as mindfulness and optimism, have been suggested to be worth investigating [12–14]. Increasing mindfulness can be an antidote for these common mental health issues because a mindless state rarely stays at the present; instead, they keep worrying about the future (anxious) or ruminating about the past (depressed), thereby feeling overwhelmed (stressed). Research has shown that people with higher mindfulness could regulate their emotions more effectively, resulting in increased productivity and healthy relationship than those with lower mindfulness [15, 16]. For example, fear of COVID-19 was associated with higher anxiety, depression, and stress among the Dutch and Belgian samples, but this relationship was weakened for those who scored higher in mindfulness, optimism, and resilience [12]. Another study indicated that dispositional mindfulness was the strongest predictor of psychological distress among 6412 Italian participants [13].

The COVID-19 virus was first detected in Bangladesh in early March 2020 [17]. In response, the government prohibited mass gatherings, implemented partial movement restrictions, and shut down academic institutions. During the pandemic, three studies investigated stress, anxiety, and depression among Bangladeshi adults using the same assessment tool [DASS 18, 21]. The first study was conducted in April [18], the second study in May [19], and the third study in June 2020 [20]. All three studies reported relatively high and nearly similar prevalence of those three comorbid mental health conditions. For example, the prevalence of moderate to severe stress symptoms was 38% in April, 32% in May, and 32.5% in June 2020. The prevalence of moderate to severe anxiety symptoms was 73% in April, 26% in May, and 46% in June 2020, and depressive symptoms were 49% in April, 43% in May, and 47.2% in June 2020.

Although early studies during the pandemic indicate increased anxiety, depression, and stress among Bangladeshi adults, it would be worthwhile to know if those conditions persisted one year on even when COVID-19 vaccines were available. People have also become more knowledgeable about the virus, which may help them become more resilient during the pandemic. Hence, finding factors that might protect people from being psychologically affected would be useful. Previous studies conducted on the Bangladeshi population did not adequately address this issue. First, only one study had a small proportion (3.32%) of the comparative sample who tested COVID-19 positive [18]. Second, the advent of the vaccine was a breakthrough that seemingly affected the mental health of the individuals eagerly waiting for it. Previous studies could not show the possible impact of vaccination on mental health conditions. Third, participants with a history of psychiatric illness were at the most risk for serious mental health consequences [11]. In Bangladesh, around 16.1% of the population reported having mental health issues before the pandemic [21, 22]. The reported prevalence of mental health disorders varied from 6.5% to 31.0% among adults and 13.4% to 22.9% among children [23]. Earlier studies did not account for this crucial health factor while exploring current mental health presumably attributable to the pandemic. Fourth, although previous studies examined the association of a range of nonmodifiable factors (e.g., social-demographic, health, and behavioural), no research on the Bangladeshi sample has investigated if modifiable factors such as dispositional mindfulness and optimism would link with standard mental health parameters.

## The current study

We recruited a large sample from all administrative divisions of Bangladesh proportionately matched with the division-wise confirmed COVID-19 cases [17]. We had participants who recovered from COVID-19; family members tested positive and received the vaccine. We recorded if participants had a history of mental illness before the pandemic. Finally, we

assessed our participants' mindfulness and optimism to see if these modifiable factors were associated with the outcome variables. We had two specific objectives. First, to explore the anxiety, depression, and stress level among the recruited sample. Second, to identify the most crucial modifiable and nonmodifiable factors associated with those mental health parameters. We framed several hypotheses in line with the results reported in previous studies. First, there will be a high prevalence of anxiety, depression, and stress among the sample. Second, nonmodifiable factors, such as those living in the country's Northern region (Rajshahi and Mymensingh division) and dwelling in the rural areas, would suffer worse mental health. During data collection, the Delta variant (B.1.617.2) reportedly originated in neighbouring India affected the Northern region severely as it shares a border with the Indian State of West Bengal. Third, participants with higher mindfulness and optimism and those who took both vaccine doses and had no history of mental illness would report better mental health.

## Method

### Design

Due to movement restrictions, data was collected online from May 27 to June 26, 2021. We followed the Checklist for Reporting Results of Internet ESurveys (CHERRIES) guidelines [24] for conducting and reporting this study. We used the Qualtrics TM survey tool for data collection [25]. The survey questionnaire was pretested with a small group to check errors or difficulties in understanding. The questionnaire had eight sections (see *Measures*), presented randomly to each participant. The quality of the responses was checked using the expert review system of Qualtrics Software. All responses passed the quality check by ensuring a good completion rate, no speed detection, and an excellent total completion rate [25]. Each user was enabled to access the survey link only one time. Further, to detect potential duplicates, we checked the IP address of each response and found no duplication.

### Sample recruitment

We distributed the questionnaire through social media platforms (Facebook, LinkedIn, WhatsApp) and emails. A large number of individuals ($N = 2356$) opened the questionnaire weblink. Twelve of them did not consent, and four were just previewed. Of the remaining responses, 201 were complete non-response (i.e., unit non-response), making a completion rate of 91.41%. However, 237 participants responded to only five or fewer questions from the 19 vital questions in the questionnaire, making them unusable. Responses of another three participants were discarded because they were under-aged (<18 years). Two participants identified themselves as the third gender. Due to a highly unbalanced gender ratio, we excluded these two participants. In total, 443 responses were removed, leaving a clean sample of 1897 individuals.

We estimated that 327 participants would be required to detect an $R^2$ value of 0.1, with 90% statistical power and a 5% significance level based on the inclusion of 30 explanatory variables in a multiple linear regression model [26].

### Sample description

Table 1 displays the characteristics of our sample. The sample comprised nearly an equal proportion of men (51%) and women (49%). The participants' average age was 30.25 years, with a standard deviation of 10.80 years. The mean socioeconomic positioning was 5.09 (SD = 1.53) on a ladder of 1 (lower) to 10 (upper). The participants comprised individuals from eight administrative divisions of Bangladesh. The proportion of participants from each division was

**Table 1. Sample characteristics (N = 1897).**

| Characteristics | Frequency | Percentage |
|---|---:|---:|
| Sex | | |
| Male | 968 | 51.03 |
| Female | 929 | 48.97 |
| Age (Mean, SD) | 30.25 | 10.80 |
| Socioeconomic status (Mean, SD) | 5.09 | 1.53 |
| Educational attainment | | |
| No schooling | 80 | 4.22 |
| SSC/Equivalent | 127 | 6.69 |
| HSC/Equivalent | 532 | 28.04 |
| Bachelor/Equivalent | 491 | 25.88 |
| Postgraduate | 667 | 35.16 |
| History of mental illness | | |
| No | 1611 | 84.92 |
| Yes | 221 | 11.65 |
| Prefer not to say | 65 | 3.43 |
| Division | | |
| Dhaka | 840 | 44.28 |
| Barisal | 122 | 6.43 |
| Rajshahi | 142 | 7.49 |
| Rangpur | 116 | 6.11 |
| Sylhet | 111 | 5.85 |
| Khulna | 107 | 5.64 |
| Chattogram | 359 | 18.92 |
| Mymensingh | 100 | 5.27 |
| Living place | | |
| Rural area | 472 | 24.88 |
| Urban area | 1425 | 75.12 |
| Current working status | | |
| Paid service | 630 | 33.21 |
| Business/Farming | 189 | 9.96 |
| Nonpaid services | 433 | 22.83 |
| Students | 645 | 34.00 |
| Frontliner | | |
| No | 1520 | 80.13 |
| Yes | 377 | 19.87 |
| Marital status | | |
| Unmarried | 1077 | 56.77 |
| Married | 820 | 43.23 |
| COVID-19 infection (self) | | |
| No | 1646 | 86.77 |
| Yes | 251 | 13.23 |
| Treatment for Covid 19 done (n = 243) | | |
| Home | 216 | 88.89 |
| Hospital | 27 | 11.11 |
| COVID-19 infection (Family member) | | |
| No | 1062 | 55.98 |
| Yes | 835 | 44.02 |

(*Continued*)

**Table 1.** (Continued)

| Characteristics | Frequency | Percentage |
|---|---|---|
| Death of family/relatives for Covid 19 (*n* = 801) | | |
| *No* | 516 | 64.42 |
| *Yes* | 285 | 35.58 |
| Vaccination status | | |
| *One doze* | 59 | 3.11 |
| *Two dozes* | 289 | 15.23 |
| *No* | 1549 | 81.66 |
| Willingness to take vaccine (*n* = 1500) | | |
| *Yes* | 870 | 58.00 |
| *No* | 291 | 19.40 |
| *Don't know* | 339 | 22.60 |
| Anxiety score (Mean, SD) | 20.94 | 13.33 |
| Anxiety category | | |
| *None (0–7)* | 371 | 19.6 |
| *Mild (8–15)* | 341 | 18.0 |
| *Moderate (16–25)* | 463 | 24.4 |
| *Severe (26–63)* | 722 | 38.1 |
| Depression score (Mean, SD) | 9.30 | 6.69 |
| Depression Category | | |
| *None (0–4)* | 516 | 27.2 |
| *Mild (5–9)* | 521 | 27.5 |
| *Moderate (10–14)* | 435 | 22.9 |
| *Moderately severe (15–19)* | 257 | 13.5 |
| *Severe (20–27)* | 168 | 8.9 |
| Stress score (Mean, SD) | 8.00 | 3.18 |
| Optimism (Mean, SD) | 7.95 | 2.29 |
| Mindfulness (Mean, SD) | 17.60 | 5.73 |

nearly similar to confirmed COVID-19 cases ($\chi^2$ = 2.65, *p* = .23, Fig 1). A quarter of the participants lived in rural areas.

Most of our participants had a formal education, of which more than half (61%) completed a bachelor's or master's degree. Nearly 12% of the participants (*n* = 211) had a history of mental illness before the pandemic and one-third (34%) worked in different paid services, and one-third (34%) were current students. The rest were either involved in nonpaid services (22%) or doing business/farming (10%). Around 20% of the participants served as COVID-19 frontliners. Roughly 13% of participants recovered from COVID-19. Around 18% (*n* = 348) received the COVID-19 vaccine, of which 15.2% (*n* = 289) had completed two doses, and 3.1% (*n* = 59) had only one dose. Of those who did not take a vaccine, 58% expressed interest in taking it when the opportunity came, while 20% were not interested in vaccination.

## Ethical consideration

Participation in this study was voluntary. The protocol was approved by the Research Ethics Committee of the Department of Educational and Counselling Psychology of Dhaka University (DECP/09/22). In addition, we presented a clear explanatory statement to each participant to make an informed decision on her/his participation. Specifically, we mentioned the risk and benefits, confidentially, and participants' right to withdraw from the study without any

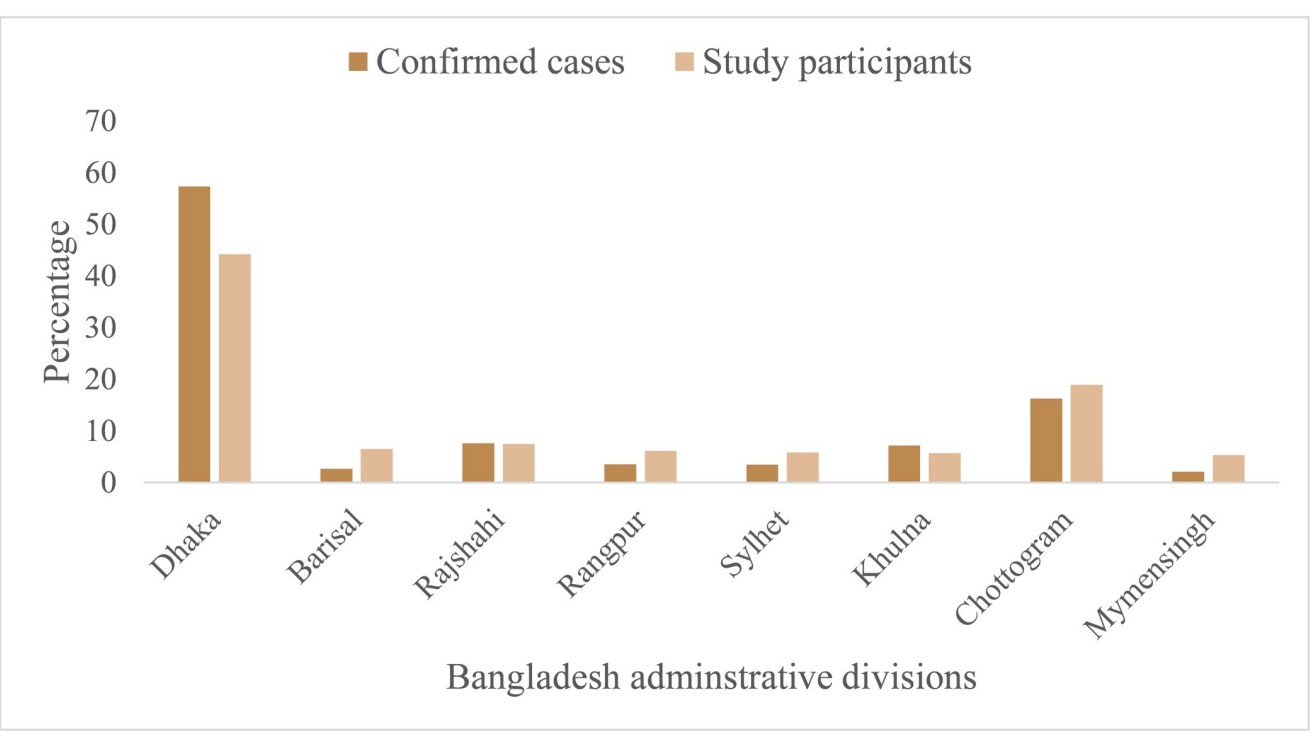

**Fig 1. The proportion of confirmed COVID-19 cases and study participants across eight administrative divisions.**

justification. Respondents consented by clicking the "Yes, I agree to participate" button. However, we offered a small prize (BDT 500 each) to ten raffle draw winners.

## Measures

**Sociodemographic and COVID-19-related factors.** We recorded participants' age (in years), sex (male, female, third gender), marital status, level of education (no schooling, up to Secondary School Certificate –10 years of schooling, up to Higher Secondary Certificate –12 years of schooling, up to Bachelor and Postgraduation), administrative divisions they live in (Dhaka, Barisal, Rajshahi, Rangpur, Sylhet, Khulna, Chattogram and Mymensingh), living area (urban or rural), and profession (paid services, business/farming, nonpaid services, and students). We also recorded their subjective socioeconomic status [SES, 27] using a ladder containing 10 blocks (1 = lowest SES, 10 = highest SES), if they had a history of mental illness, recovered from COVID-19 infection, and if their family members were infected. They also shared if they had taken the COVID-19 vaccine.

**Psychological factors.** *Optimism.* We measured optimism using the Revised Life Orientation Test [LOT-R, 28]. The LOT-R consists of 10 items, three items assess optimism (example item: In uncertain times, I usually expect the best), three-item assess pessimism (example item: I hardly ever expect things to go my way), and four items are fillers (example item: It is easy for me to relax). The fillers were not counted in calculating the scores. Participants were asked to indicate their level of agreement with each item on a 5-point Likert scale (0 = Strongly disagree to 4 = Strongly agree). The LOT-R can be scored in two ways: as a summative index of dispositional optimism by combining scores of all items (i.e., pessimism items need to be reversed coded) or as an optimism and pessimism scale [14]. In this study, we only used optimism scores.

*Mindfulness*. We assessed dispositional mindfulness using the five-item Mindful Attention Awareness Scale (MAAS-5) version. The original MAAS was a 15-item single-dimension tool to measure the frequency of open and receptive attention to and awareness of ongoing events and experiences [29, 30]. Recently, a shorter version has been used, with equally good psychometric properties [31, 32]. We prefer to use the shorter version to lessen the burden on the participants' shoulders [32]. Items are presented using a 6-point Likert scale ranging from 1 (almost always) to 6 (almost never). The total mindfulness score was calculated by adding all item scores. A higher score indicates a higher mindfulness. Internal consistency reliability of the MAAS-5 for the current sample was very good (Cronbach's alpha .817).

**Outcome measures.** *Anxiety*. We used the 21-item Beck Anxiety Inventory [BAI, 33] to measure the symptoms and severity of anxiety. The BAI was initially developed for clinical samples, but later it was found efficient in assessing anxiety symptoms among the non-clinical samples [34]. The items were developed from anxiety disorder symptoms outlined in the Diagnostic and Statistical Manual of Mental Disorders [DSM V, 35]. Participants were asked to report to what extent they had been bothered by each of the symptoms over the past week on a 4-point Likert scale ranging from '0' (not at all) to '3' (I could barely stand it). The total raw score is calculated by summing up the 21 items, giving a total score ranging from 0 to 63, where a higher score indicates a higher anxiety level. Following the scoring manual, we transformed raw scores into four severity categories (0–7: minimal anxiety, 8–15: mild anxiety, 16–25: moderate anxiety, and 26–63: severe anxiety) [36, 37]. The Bangla BAI [38] for the current sample showed excellent internal consistency (Cronbach's alpha .934).

*Depression*. We used the Bangla-validated 9-item Patient Health Questionnaire [PHQ-9, 39, 40] to measure depression [41]. Participants responded to what extent they were bothered by each of the nine depressive symptoms over the last two weeks on a scale from '0' (not at all) to '3' (nearly every day). The total score was obtained by adding all item scores, ranging from 0 to 27, where a higher score indicates a higher level of depression. We used continuous scores and categories of depression severity (1–4: None; 4–9: Mild; 10–14: Moderate; 15–19: Moderately severe; 20–27). The internal consistency reliability of the PHQ-9 for the current sample was very good (Cronbach's alpha .872).

*Stress*. We used the 4-item version of the Perceived Stress Scale [PSS-4, 42]], where respondents rated how often they have experienced stress over the last two weeks on a 5- point scale ranging from '0'(Never) to '4' (Very often). Total PSS-4 scores ranged from 0 to 16, with higher scores indicating higher stress levels. The scale's internal consistency for the current sample was moderate (Cronbach's alpha .563).

## Statistical analysis

While cleaning the data set, we noticed less than 5% missing values. We performed multiple imputations to get a cleaner data set. Sensitivity analysis showed no significant difference between the original and imputed datasets.

We present the mean with SD of anxiety, depression, and stress scores for the whole sample to denote the current state of those conditions. Differences in those conditions among groups were tested either by independent sample t-test (for two groups) or one-way Analysis of Variance (for more than two groups).

To identify the most crucial modifiable and nonmodifiable factors associated with anxiety, depression, and stress, we conducted univariate and multivariate regression analyses incorporating all modifiable and nonmodifiable factors. Results of univariate regression models have been reported in S1 File. Serving as a COVID-19 frontliner and COVID-19 infection status (self) did not affect any of the outcomes in the univariable regression; therefore, they were

excluded from the multivariate analyses. We ran two models for each outcome. In the first model, we entered only nonmodifiable factors (age, sex, education, division, living area, current job, and marital status) as predictors. In the second model, we added modifiable predictors, such as a history of mental illness, subjective socioeconomic status, family member's COVID-19 infection, self-vaccination status, optimism, and mindfulness.

Categorical predictors with more than two categories were dummy coded. In this process, four dummy variables were created for education (reference category: postgraduation), two variables for the history of mental illness (reference category: no history), seven variables for division (reference category: Dhaka), three variables for the current job (reference category: paid services) and two for vaccination status (reference category: no vaccination).

Assumptions of linear regression (normality and homoscedasticity of residuals, linearity with continuous variables) and the presence of outliers were assessed graphically. The Mahalanobis distance indicates around 57 cases that can be considered multivariate outliers. We conducted the analyses after discarding those influential cases. Results showed no difference from the original models obtained with all participants. Therefore, all statistical analyses were performed with the complete set of data. Multicollinearity was checked using variance inflation factors (VIF) and found to be acceptable levels. We did square root transformations for depression and anxiety scores to satisfy assumptions. The robustness of the models was examined by removing several cases with large residuals ($< -3$ or $> 3$) and comparing results to the original models. There was no difference in the revised models.

## Results

### Level of anxiety, depression, and stress among the study sample

The mean anxiety score for the total sample was 20.94 (SD 13.33), depression 9.30 (SD 6.69), and stress 8.00 (SD 3.18). Based on the established cut-off points, around 62% of the participants suffered moderate to severe clinical anxiety symptoms and 45.3% from clinical depression (Table 1). Group-wise comparisons of these conditions are presented in Table 2. Women reported significantly higher anxiety ($t$ = -3.82, $p$ < .0001, $d$ = 0.18) and stress ($t$ = -3.29, $p$ = .001, $d$ = 0.15) than men did. Depression scores between the two sexes did not vary significantly ($p$ = .077). Participants with higher secondary education appeared to be the most vulnerable group. Compared to participants holding postgraduate education, this group reported significantly higher anxiety (*Mean difference* = 3.67, $p$ < .0001), depression (*Mean difference* = 3.03, $p$ < .0001), and stress (*Mean difference* = 1.08, $p$ < .0001) symptoms. The remaining groups were not different in terms of mental health outcomes. The mental health outcomes of those with a history of mental illness were poor compared to participants without a history of mental illness.

Among the eight divisions, participants from Rajshahi and Mymensingh reported the highest level of anxiety and depression, while participants from Rangpur and Chattogram reported the lowest level of anxiety and depression. Perceived stress did not vary significantly among participants from various divisions. Compared to their urban counterparts, participants in rural areas reported significantly higher anxiety ($t$ = 3.93, $p$ < .0001, $d$ = 0.20). However, these two groups did not vary in depression and stress ($p$>.05).

We found that current students and those in nonpaid services reported a higher level of anxiety, depression, and stress than participants involved in paid services or business/farming. COVID-19 frontliners experienced significantly higher stress than those who were not ($t$ = 4.13, $p$ < .0001, $d$ = 0.25). However, anxiety and depression scores were not significantly different between these two groups ($p$>.05). Unmarried, as opposed to married participants, scored poorly in all three mental health measures.

**Table 2. Anxiety, depression, and stress of participants according to demographic variables and COVID-19 vaccination status (*N* = 1897).**

| Variables | Anxiety, depression, and stress status | | | | | |
|---|---|---|---|---|---|---|
| | Anxiety (Mean, SD) | *t/F*-value | Depression (Mean, SD) | *t/F*-value | Stress (Mean, SD) | *t/F*-value |
| **Sex** | | | | | | |
| Male | 19.80 (12.78) | -3.82*** | 9.04 (6.50) | -1.77 | 7.77 (3.12) | -3.29*** |
| Female | 22.13 (13.78) | | 9.58 (6.87) | | 8.25 (3.23) | |
| **Education** | | | | | | |
| No schooling | 20.33 (14.11) | 6.95*** | 8.33 (6.80) | 18.02 | 8.31 (1.87) | 9.19 |
| SSC/Equivalent | 20.07 (12.96) | | 8.54 (6.15) | | 7.94 (2.64) | |
| HSC/Equivalent | 22.66 (12.85) | | 10.86 (6.39) | | 8.54 (3.18) | |
| Bachelor/Equivalent | 22.07 (13.81) | | 9.97 (6.75) | | 8.12 (3.13) | |
| Postgraduate | 18.99 (13.07) | | 7.83 (6.63) | | 7.46 (3.36) | |
| **History of mental illness** | | | | | | |
| No | 19.92 (12.97) | 32.63*** | 8.76 (6.50) | 35.76*** | 7.84 (3.17) | 13.50*** |
| Yes | 27.04 (13.82) | | 12.37 (6.79) | | 8.83 (3.14) | |
| Prefer Not to Say | 25.46 (14.11) | | 12.18 (7.41) | | 9.11 (3.16) | |
| **Division** | | | | | | |
| Dhaka | 21.61 (13.32) | 29.79*** | 10.03 (6.95) | 22.56*** | 7.96 (3.60) | 1.03 |
| Barisal | 23.29 (12.15) | | 9.92 (5.75) | | 7.91 (2.87) | |
| Rajshahi | 26.02 (11.79) | | 12.49 (5.78) | | 8.41 (2.85) | |
| Rangpur | 18.92 (12.52) | | 8.84 (6.64) | | 7.69 (3.52) | |
| Sylhet | 24.44 (10.77) | | 9.43 (5.20) | | 8.10 (2.55) | |
| Khulna | 26.71 (10.68) | | 9.87 (5.43) | | 8.53 (2.60) | |
| Chattogram | 13.21 (13.52) | | 5.79 (6.54) | | 7.92 (2.71) | |
| Mymensingh | 25.33 (9.33) | | 10.35 (5.14) | | 7.85 (2.55) | |
| **Living place** | | | | | | |
| Rural | 22.90 (12.08) | 3.93*** | 9.73 (5.94) | 1.73 | 7.91 (2.84) | -0.78 |
| Urban | 20.29 (13.66) | | 9.16 (6.91) | | 8.03 (3.29) | |
| **Current job** | | | | | | |
| Paid services | 19.36 (12.40) | 5.61** | 7.51 (5.81) | 30.15*** | 7.26 (3.17) | 19.53*** |
| Business/Farming | 20.35 (13.03) | | 8.80 (6.08) | | 7.88 (2.64) | |
| Nonpaid services | 21.47 (14.52) | | 9.68 (7.30) | | 8.48 (2.97) | |
| Student | 22.31 (13.30) | | 10.95 (6.79) | | 8.44 (3.34) | |
| **Frontliner** | | | | | | |
| No | 20.92 (13.48) | -0.16 | 9.35 (6.79) | 0.58 | 8.16 (3.13) | 4.13*** |
| Yes | 21.04 (12.71) | | 9.12 (6.25) | | 7.38 (3.33) | |
| **Marital status** | | | | | | |
| Unmarried | 22.26 (13.02) | 4.92*** | 10.41 (6.57) | 8.44*** | 8.31 (3.29) | 4.95*** |
| Married | 19.22 (13.54) | | 7.84 (6.57) | | 7.60 (2.98) | |
| **COVID-19 infection (self)** | | | | | | |
| No | 20.72 (13.13) | -1.78 | 9.24 (6.62) | -1.05 | 8.00 (3.13) | -0.11 |
| Yes | 22.44 (14.46) | | 9.71 (7.13) | | 8.02 (3.50) | |
| **COVID-19 infection (Family member)** | | | | | | |
| No | 20.22 (12.98) | -2.68 | 8.75 (6.35) | -4.05*** | 7.88 (3.09) | -1.84 |
| Yes | 21.87 (13.71) | | 10.00 (7.04) | | 8.16 (3.29) | |
| **Vaccination status** | | | | | | |

*(Continued)*

**Table 2.** (Continued)

| Variables | Anxiety, depression, and stress status | | | | | |
|---|---|---|---|---|---|---|
| | Anxiety (Mean, SD) | t/F-value | Depression (Mean, SD) | t/F-value | Stress (Mean, SD) | t/F-value |
| No | 22.44 (12.94) | 58.15*** | 10.08 (6.49) | 62.10*** | 8.13 (3.20) | 7.43** |
| Yes, one doze | 16.93 (12.79) | | 7.27 (6.54) | | 7.81 (2.86) | |
| Yes, two dozes | 13.73 (12.96) | | 5.56 (6.41) | | 7.35 (3.09) | |

*$p < .05$,

**$p < .01$,

***$p < .0001$

*Note*. Test statistic was obtained through t-tests (for two groups) or ANOVA (for more than two groups).

Surprisingly, recovered COVID-19 patients did not differ from the non-infected participants in any outcome measures. Nonetheless, participants who had family members tested positive were more vulnerable in terms of anxiety ($t = -2.66$, $p = .007$, $d = 0.12$) and depression ($t = -4.01$, $p < .0001$, $d = 0.19$) than those whose family members remained unaffected. The difference in stress between these two groups did not reach statistical significance ($p = .065$).

The mental health outcomes of participants who were not vaccinated and took single and double doses differed significantly. For example, compared to participants without vaccination, participants with a double dose had significantly lower anxiety (*Mean difference* = 8.72, $p < .0001$), depression (*Mean difference* = 5.52, $p < .0001$), and stress (*Mean difference* = 0.78, $p < .0001$). Similarly, compared to participants without vaccination, participants with single-dose vaccination also had significantly lower anxiety (*Mean difference* = 5.51, $p = .004$) and depression (*Mean difference* = 2.81, $p = .003$). However, the single-dose vaccination did not impact perceived stress ($p = .731$). Those who had a single dose and double doses were not different in any mental health outcomes.

## Modifiable and nonmodifiable factors associated with anxiety

We report the results of multivariate regression analyses in Tables 3–5. For anxiety, nonmodifiable predictors jointly accounted for 19% of the variances, with sex, age, education, division, and living place emerging as significant predictors (Model 1). The second model incorporated modifiable predictors, accounting for 33% of the variance in the anxiety scores, a significant increase of 14% attributable to modifiable factors. Female gender ($B = 0.31$, 95% *CI*: 0.17, 0.46), secondary education ($B = -0.31$, 95% *CI*: -0.61, -0.01), division (Rajshahi [$B = 0.38$, 95% *CI*: 0.11, 0.65], Sylhet [$B = 0.41$, 95% *CI*: 0.11, 0.71], Khulna [$B = 0.45$, 95% *CI*: 0.14, 0.76], Chattogram [$B = -1.20$, 95% *CI*: -1.39, -1.01], Mymensingh [$B = 0.46$, 95% *CI*: 0.15, 0.77]) and living in urban ($B = -0.31$, 95% *CI*: -0.49, -0.14), nonpaid services ($B = -0.28$, 95% *CI*: -0.48, -0.09), history of mental illness (yes [$B = 0.61$, 95% *CI*: 0.40, 0.82], prefer not to say [$B = 0.60$, 95% *CI*: 0.24, 0.96]), socioeconomic status ($B = -0.07$, 95% *CI*: -0.11, -0.02), family member tested positive ($B = 0.28$, 95% *CI*: 0.14, 0.42), completion of double doses of vaccine ($B = -0.66$, 95% *CI*: -0.87, -0.44), optimism ($B = -0.10$, 95% *CI*: -0.13, -0.07), and mindfulness ($B = -0.08$, 95% *CI*: -0.09, -0.07) emerged as significant and independent predictors of anxiety in the fully adjusted model. Among them, mindfulness ($\beta = -.26$), living in Chattogram division ($\beta = -.26$), optimism ($\beta = -.13$), completion of a double dose of vaccine ($\beta = -.13$), and history of mental illness ($\beta = .11$) were the strongest predictors of anxiety. Except for division, the remaining four crucial predictors are modifiable.

**Table 3. Nonmodifiable and modifiable factors associated with anxiety among Bangladeshi adults (N = 1897).**

| Explanatory variables | Model 1 | | | Model 2 | | |
|---|---|---|---|---|---|---|
| | *B [95% CI]* | *SE* | *β* | *B [95% CI]* | *SE* | *β* |
| **Sex** | | | | | | |
| *Male (ref)* | | | | | | |
| *Female* | 0.34*** (0.18, 0.49) | .08 | .09 | 0.31*** (0.17, 0.46) | .07 | .09 |
| Age (in years) | -0.02*** (-0.03, -0.01) | .00 | -.13 | -0.01* (-.02, -0.00) | .00 | -.06 |
| **Education** | | | | | | |
| *Postgraduate (ref)* | | | | | | |
| *Nonformal* | 0.28 (-0.14, 0.69) | .21 | .03 | -0.03 (-0.41, 0.45) | .19 | .00 |
| *SSC/Equivalent* | -0.06 (-0.38, 0.27) | .17 | -.01 | -0.31* (-0.61, -0.01) | .15 | -.04 |
| *HSC/Equivalent* | 0.21* (-0.02, 0.43) | .12 | .05 | 0.02 (-0.19, 0.23) | .11 | .00 |
| *Bachelor/Equivalent* | 0.25* (0.04, 0.45) | .10 | .06 | 0.07 (-0.12, 0.26) | .10 | .02 |
| **Division** | | | | | | |
| *Dhaka (ref)* | | | | | | |
| *Barisal* | 0.09 (-0.22, 0.40) | .16 | .01 | 0.23 (-0.06, 0.51) | .15 | .03 |
| *Rajshahi* | 0.47*** (0.17, 0.77) | .15 | .07 | 0.38** (0.11, 0.65) | .14 | .06 |
| *Rangpur* | -0.47** (-0.79, -0.14) | .17 | -.06 | -0.26 (-0.56, 0.04) | .15 | -.03 |
| *Sylhet* | 0.38* (0.05, 0.71) | .17 | .05 | 0.41** (0.11, 0.71) | .15 | .05 |
| *Khulna* | 0.41* (0.07, 0.75) | .17 | .05 | 0.45*** (0.14, 0.76) | .16 | .06 |
| *Chattogram* | -1.51*** (-1.71, -1.31) | .10 | -.33 | -1.20*** (-1.39, -1.01) | .10 | -.26 |
| *Mymensingh* | 0.47* (0.13, 0.81) | .17 | .06 | 0.46*** (0.15, 0.77) | .16 | .06 |
| **Living place** | | | | | | |
| *Rural (ref)* | | | | | | |
| *Urban* | -0.38*** (-0.56, -0.19) | .10 | -.09 | -.31* (-0.49, -0.14) | .09 | -.08 |
| **Job** | | | | | | |
| *Paid services (ref)* | | | | | | |
| *Business/Farming* | -0.04 (-0.24, 0.33) | .14 | .01 | -0.09 (-0.35, 0.16) | .13 | -.02 |
| *Nonpaid services* | 0.00 (0.21, 0.21) | .11 | .00 | -0.28*** (-0.48, -0.09) | .10 | -.07 |
| *Student* | -0.11 (-0.35, 0.12) | .12 | -.05 | -0.21 (-0.42, 0.01) | .11 | -.06 |
| **Marital status** | | | | | | |
| *Unmarried (ref)* | | | | | | |
| *Married* | -0.19 (-0.39, 0.01) | .10 | -.05 | -0.14 (-0.33, 0.04) | .09 | -.04 |
| **History of mental illness** | | | | | | |
| *No (ref)* | | | | | | |
| *Yes* | | | | 0.61*** (0.40, 0.82) | .11 | .11 |
| *Prefer Not to Say* | | | | 0.60*** (0.24, 0.96) | .19 | .06 |
| Socioeconomic status | | | | -0.07*** (-0.11, -0.02) | .02 | -.06 |
| **COVID-19 Infection (Family member)** | | | | | | |
| *No (ref)* | | | | | | |
| *Yes* | | | | 0.28*** (0.14, 0.42) | .07 | .08 |
| **Vaccination status (Self)** | | | | | | |
| *No (ref)* | | | | | | |
| *Yes, one doze* | | | | -0.29 (-0.68, 0.10) | .20 | -.03 |
| *Yes, two dozes* | | | | -0.66*** (-0.87, -0.44) | .11 | -.13 |
| **Optimism** | | | | -0.10*** (-0.13, -0.07) | .02 | -.13 |
| **Mindfulness** | | | | -0.08*** (-0.09, -0.07) | .00 | -.26 |
| **Adjusted $R^2$** | .19*** | | | .33*** | | |

(Continued)

**Table 3.** (Continued)

| Explanatory variables | Model 1 | | | Model 2 | | |
|---|---|---|---|---|---|---|
| | *B* [95% CI] | SE | β | *B* [95% CI] | SE | β |
| Adjusted $R^2$ change | - | | | .14*** | | |

*Note.* Dependent variable: Anxiety scores (square root transformed),

*** $p<0.001$,

** $p<0.01$,

* $p<0.05$

Model 1: Nonmodifiable predictors (sex, age, education, division, living place, job, and marital status)

Model 2: adjusted for model 1 + modifiable predictors (history of mental illness, subjective socioeconomic status, family member's COVID-19 infection status, vaccination status, optimism, and mindfulness)

### Modifiable and nonmodifiable factors associated with depression

To determine the factors associated with depression, we replicated the models run for anxiety (Table 4). Nonmodifiable factors jointly accounted for 20% of the variances in the depression scores, with age, education, divisions and living place, current job, and marital status emerging as significant predictors (Model 1). When modifiable predictors were added to the multivariable model for depression, the model's explanatory power increased to 41%. Modifiable predictors caused an additional 21% of the variances in the depression scores. Having a secondary education (*B* = -0.27, 95% *CI*: -0.48, -0.05), division (Rajshahi [*B* = 0.32, 95% *CI*: 0.13, 0.52], Rangpur [*B* = -0.23, 95% *CI*: -0.45, -0.02], Chattogram [*B* = -0.83, 95% *CI*: -0.97, -0.69]), living in urban (*B* = -0.14, 95% *CI*: -0.26, -0.02), business/farming (*B* = 0.19, 95% *CI*: 0.01, 0.37), being married (*B* = -0.18, 95% *CI*: -0.31, -0.05), history of mental illness (yes [*B* = 0.43, 95% *CI*: 0.29, 0.58], prefer not to say [*B* = 0.45, 95% *CI*: 0.20, 0.71]), socioeconomic status (*B* = -0.05, 95% *CI*: -0.08, -0.02), family member tested positive (*B* = 0.24, 95% *CI*: 0.14, 0.34), completion of double doses of vaccine (*B* = -0.39, 95% *CI*: -0.54, -0.24), optimism (*B* = -0.09, 95% *CI*: -0.11, -0.07), and mindfulness (*B* = -0.08, 95% *CI*: -0.09, -0.07)-emerged as significant and independent predictors of depression in the fully adjusted model. Based on the unique contribution, mindfulness (β = -.35), living in Chattogram division (β = -.24), optimism (β = -.15), age (β = -.12), completion of a double dose of vaccine (β = -.10), and having a history of mental illness (β = .10) were found the strongest predictors of depression.

### Modifiable and nonmodifiable factors associated with stress

Participants living places (urban/rural or division) do not affect stress scores in the univariable regression; therefore, these factors were excluded from the multivariate analyses. Results indicated that nonmodifiable predictors jointly accounted for only 4% of the variances in the stress scores, with female gender, education, current job, and marital status emerging as significant predictors (Model 1). The second model accounted for 29% of the variances in the stress scores, a significant increase of 25% variances attributable to modifiable factors. In the fully adjusted model, female gender (*B* = -0.47, 95% *CI*: 0.21, 0.73), current job (business/farming [*B* = 0.48, 95% *CI*: 0.00, 0.95], nonpaid services [*B* = 0.45, 95% *CI*: 0.09, 0.80]), being married (*B* = -0.54, 95% *CI*: -0.87, -0.21), socioeconomic status (*B* = -0.23, 95% *CI*: -0.31, -0.15), family member tested positive (*B* = 0.27, 95% *CI*: 0.02, 0.52), optimism (*B* = -0.34, 95% *CI*: -0.40, -0.29), and mindfulness (*B* = -0.20, 95% *CI*: -0.22, -0.18)-emerged as significant and independent predictors of stress. For stress, mindfulness (β = -.36), optimism (β = -.25), socioeconomic status (β = -.11), being married (β = -.08), and being female (β = .07) were found to be the strongest predictors.

**Table 4. Nonmodifiable and modifiable factors associated with depression among Bangladeshi adults (*N* = 1897).**

| Explanatory variables | Model 1 | | | Model 2 | | |
|---|---|---|---|---|---|---|
| | *B* [95% CI] | *SE* | *β* | *B* [95% CI] | *SE* | *β* |
| **Sex** | | | | | | |
| *Male (ref)* | | | | | | |
| *Female* | 0.11 (-0.01, 0.22) | .06 | .04 | 0.10 (-0.00, 0.20) | .05 | .04 |
| **Age (in years)** | -0.02*** (-0.03, -0.01) | .00 | -.18 | -0.01*** (-0.02, -0.01) | .00 | -.12 |
| **Education** | | | | | | |
| *Postgraduate (ref)* | | | | | | |
| *Nonformal* | 0.21 (-0.10, 0.52) | .16 | .03 | -0.07 (-0.34, 0.20) | .14 | -.01 |
| *SSC/Equivalent* | -0.03 (-0.27, 0.21) | .12 | -.01 | -0.27* (-0.48, -0.05) | .11 | -.05 |
| *HSC/Equivalent* | 0.28*** (0.11, 0.45) | .09 | .09 | 0.11 (-0.04, 0.26) | .08 | .04 |
| *Bachelor/Equivalent* | 0.26*** (0.11, 0.41) | .08 | .08 | 0.10 (-0.03, 0.24) | .07 | .03 |
| **Division** | | | | | | |
| *Dhaka (ref)* | | | | | | |
| *Barisal* | -0.06 (-0.29, 0.18) | .12 | -.01 | 0.06 (-0.15, 0.26) | .10 | .01 |
| *Rajshahi* | 0.41*** (0.18, 0.63) | .11 | .08 | 0.32*** (0.13, 0.52) | .10 | .06 |
| *Rangpur* | -0.42*** (-0.67, -0.18) | .12 | -.07 | -0.23* (-0.45, -0.02) | .11 | -.04 |
| *Sylhet* | 0.00 (-0.24, 0.25) | .13 | .00 | 0.03 (-0.19, 0.25) | .11 | .01 |
| *Khulna* | -0.11 (-0.36, 0.15) | .13 | -.02 | -0.08 (-0.30, 0.14) | .11 | -.01 |
| *Chattogram* | -1.08*** (-1.24, -0.93) | .08 | -.31 | -0.83*** (-0.97, -0.69) | .07 | -.24 |
| *Mymensingh* | 0.06 (-0.20, 0.32) | .13 | .01 | 0.05 (-0.18, 0.27) | .11 | .01 |
| **Living place** | | | | | | |
| *Rural (ref)* | | | | | | |
| *Urban* | -0.19*** (-0.33, -0.05) | .07 | -.06 | -0.14* (-0.26, -0.02) | .06 | -.04 |
| **Job** | | | | | | |
| *Paid services (ref)* | | | | | | |
| *Business/Farming* | 0.29** (0.08, 0.50) | .11 | .07 | 0.19* (0.01, 0.37) | .09 | .04 |
| *Nonpaid services* | 0.26*** (0.11, 0.42) | .08 | .08 | 0.02 (-0.12, 0.16) | .07 | .01 |
| *Student* | 0.14 (-0.04, 0.32) | .09 | .05 | 0.06 (-0.10, 0.21) | .08 | .02 |
| **Marital status** | | | | | | |
| *Unmarried (ref)* | | | | | | |
| *Married* | -0.20* (-0.35, -0.05) | .08 | -.07 | -0.18** (-0.31, -.05) | .07 | -.06 |
| **History of mental illness** | | | | | | |
| *No (ref)* | | | | | | |
| *Yes* | | | | 0.43*** (0.29, 0.58) | .08 | .10 |
| *Prefer Not to Say* | | | | 0.45*** (0.20, 0.71) | .13 | .06 |
| **Socioeconomic status** | | | | -0.05*** (-0.08, -0.02) | .02 | -.06 |
| **COVID-19 Infection (Family member)** | | | | | | |
| *No (ref)* | | | | | | |
| *Yes* | | | | 0.24*** (0.14, 0.34) | .05 | .09 |
| **Vaccination status (Self)** | | | | | | |
| *No (ref)* | | | | | | |
| *Yes, one doze* | | | | -0.12 (-0.40, 0.16) | .14 | -.02 |
| *Yes, two dozes* | | | | -0.39*** (-0.54, -0.24) | .08 | -.10 |
| **Optimism** | | | | -0.09*** (-0.11, -0.07) | .01 | -.15 |
| **Mindfulness** | | | | -0.08*** (-0.09, -0.07) | .00 | -.35 |
| **Adjusted $R^2$** | .20 | | | .41 | | |

(*Continued*)

**Table 4.** (Continued)

| Explanatory variables | Model 1 | | | Model 2 | | |
|---|---|---|---|---|---|---|
| | B [95% CI] | SE | β | B [95% CI] | SE | β |
| Adjusted $R^2$ change | - | | | .21 | | |

*Note*. Dependent variable: Depression (PHQ-9) scores (square root transformed),

*** $p<0.001$,

** $p<0.01$,

* $p<0.05$

Model 1: Nonmodifiable predictors (sex, age, education, division, living place, job, and marital status)

Model 2: adjusted for model 1 + modifiable predictors (history of mental illness, subjective socioeconomic status, family member's COVID-19 infection status, vaccination status, optimism, and mindfulness)

## Discussion

This study addressed two research questions. Firstly, did the prevalence of affective disorders like depression, anxiety, and stress increase among Bangladeshi adults one year after the pandemic? And secondly, which factors—both modifiable and nonmodifiable—were associated with these three types of affective disorders? The results showed a high prevalence of depression, anxiety, and stress among the participants, which is quite similar to the rates reported in previous studies conducted in Bangladesh [18–20, 43]. However, due to the cross-sectional nature of our study, we could not conclude if mental health conditions worsened or improved in our participants.

Globally, the prevalence of affective disorders increased during the COVID-19 pandemic. Ample studies compared the prevalence of depression, anxiety, and stress between the pre-and during-lockdown period in 47 countries across Asia, Europe, and America (see Daniali, Martinussen, & Flaten, 2023 for a systematic review and meta-analysis [44]). Daniali and colleagues found that feelings of negative emotions increased globally during the pandemic, and depression had the highest elevation. In Asian countries, there was an elevation of depression and stress, but in European countries, there was an elevation of depression only. However, the negative emotions remained unchanged in American countries. In the later phase of the pandemic, the stress level dropped globally, and European countries recorded a lower level of stress and anxiety.

Santomauro and colleagues [45] conducted a large-scale systematic review of 48 articles (46 studies met the criteria for major depressive disorder and 27 for anxiety disorder) reporting the prevalence of major depressive disorder and anxiety disorder during the COVID-19 pandemic. They observed that globally, there was a 27.6% increase in depressive disorders and a 25.6% increase in anxiety disorders. While comparing the prevalence rates across world regions, they found that the South Asian region had the highest prevalence, and South-East Asia, East Asia, and the Oceanian region had the lowest prevalence of depression and anxiety disorder. The South Asian region noted a 36.1% increase in depressive disorder and a 35.1% increase in anxiety disorder during the pandemic. However, South-East Asia, East Asia, and the Oceanian region recorded 11.5% and 13.8% increases in depression and anxiety disorder, respectively.

### The most at-risk group

We found that females, unmarried, college students, unemployed, and those living in the Northern region were more at risk of depression, anxiety, and stress. These findings generally confirm the results reported in previous studies in Bangladesh [18, 19, 46–48]. Whereas

**Table 5. Nonmodifiable and modifiable factors associated with perceived stress among Bangladeshi adults (N = 1897).**

| Explanatory variables | Model 1 | | | Model 2 | | |
|---|---|---|---|---|---|---|
| | B [95% CI] | SE | β | B [95% CI] | SE | β |
| **Sex** | | | | | | |
| Male (ref) | | | | | | |
| Female | 0.43*** (0.13, 0.72) | .15 | .07 | 0.47*** (0.21, 0.73) | .13 | .07 |
| **Age (in years)** | 0.00 (-0.02, 0.02) | .01 | .00 | 0.01 (-0.01, 0.02) | .01 | .03 |
| **Education** | | | | | | |
| Postgraduate (ref) | | | | | | |
| Nonformal | 0.65 (-0.13, 1.44) | .40 | .04 | -0.21 (-0.90, 0.48) | .35 | -.01 |
| SSC/Equivalent | 0.28 (-0.35, 0.90) | .32 | .02 | -0.48 (-1.03, 0.07) | .28 | -.04 |
| HSC/Equivalent | 0.70*** (0.27, 1.13) | .22 | .10 | 0.26 (-0.12, 0.63) | .19 | .04 |
| Bachelor/Equivalent | 0.36 (-0.03, 0.76) | .20 | .05 | -0.06 (-0.40, 0.28) | .18 | -.01 |
| **Job** | | | | | | |
| Paid services (ref) | | | | | | |
| Business/Farming | 0.57*** (0.03, 1.12) | .28 | .05 | 0.48* (0.00, 0.95) | .24 | .05 |
| Nonpaid services | 0.96*** (0.55, 1.36) | .21 | .13 | 0.45* (0.09, 0.80) | .18 | .06 |
| Student | 0.49* (0.03, 0.95) | .23 | .07 | 0.38 (-0.02, 0.77) | .20 | .06 |
| **Marital status** | | | | | | |
| Unmarried (ref) | | | | | | |
| Married | -0.52** (-0.90, -0.14) | .20 | -.08 | -0.54*** (-0.87, -0.21) | .17 | -.08 |
| **History of mental illness** | | | | | | |
| No (ref) | | | | | | |
| Yes | | | | 0.45** (0.06, 0.83) | .20 | .05 |
| Prefer Not to Say | | | | 0.85** (0.18, 1.52) | .34 | .05 |
| **Socioeconomic status** | | | | -0.23*** (-0.31, -0.15) | .04 | -.11 |
| **COVID-19 Infection (Family member)** | | | | | | |
| No (ref) | | | | | | |
| Yes | | | | 0.27* (0.02, 0.52) | .13 | .04 |
| **Vaccination status (Self)** | | | | | | |
| No (ref) | | | | | | |
| Yes, one doze | | | | 0.39 (-0.33, 1.10) | .36 | .02 |
| Yes, two dozes | | | | 0.19 (-0.19, 0.58) | .20 | .02 |
| **Optimism** | | | | -0.34*** (-0.40, -0.29) | .03 | -.25 |
| **Mindfulness** | | | | -0.20*** (-0.22, -0.18) | .01 | -.36 |
| **Adjusted R²** | .04 | | | .29 | | |
| **Adjusted R² change** | - | | | .25 | | |

*Note.* Dependent variable: Perceived stress scores (PSS-4),

*** $p < 0.001$,

** $p < 0.01$,

* $p < 0.05$

Model 1: Nonmodifiable predictors (sex, age, education, job, and marital status)

Model 2: adjusted for model 1 + modifiable predictors (history of mental illness, subjective socioeconomic status, family member's COVID-19 infection status, vaccination status, optimism, and mindfulness)

women, unmarried, and unemployed were more likely to have poor mental health, it was difficult to explain why college students, a nonmodifiable factor, should also have more affective disorders during the pandemic. Globally, students were the most affected group. Like many other countries [44], Bangladesh had to shut down academic institutions for an extended

period. More specifically, the college students faced severe study disruption—their final exams did not take place and were subject to auto-promotion. This unusual situation, in which there was no delivery of the curriculum and no formal assessment, concerned many because they would have to sit for the competitive university entrance exams in the days to come. Hence, poor mental health for this group is understandable. Similar to our results, females across the world also reported a higher level of depression, anxiety, and stress than males during the pandemic, and this was more pronounced in the European region [44].

Initially, Dhaka, the central region, was the epicentre of the COVID-19 pandemic. We compared the mental health of participants from this region to those recruited from the outskirts of the country. The results showed that participants from the Northern districts had significantly poorer mental health than those from the Central region. This result reflects the pattern of the spread of COVID-19 infection in Bangladesh. During data collection, the second wave of the pandemic, characterized by the Delta variant (B.1.617.2), penetrated the Northern districts, which share a border with India, where the variant was first documented [49]. The confirmed cases and casualties increased alarmingly at that time [17]. Interestingly, despite being in the same pandemic, we found participants from the South-Eastern districts reporting better mental health than participants from the Central region. This improved mental health condition among people from the South-East needs further examination.

Typically, people living in urban areas showed poorer mental health than their rural counterparts, nowadays referred to as "urbanicity and risk of common mental disorders (CMDs)" [50]. However, for the current sample, we found that participants living in rural areas reported higher anxiety, depression, and stress than those in city areas. In the early stage, COVID-19 cases were more apparent in the megacities, but it has spread over time, affecting people in the countryside in large numbers. Additionally, the vaccine was first made available to people in the urban areas, keeping the rural people waiting. There were more awareness campaigns within the big cities, which might have positively affected the mental health of the urban citizen.

## Modifiable factors affecting anxiety, depression, and stress

We have found several modifiable determinants in our sample that together accounted for an additional 10%-25% of the variances in anxiety, depression, and stress after adjusting for the effect of nonmodifiable factors. Consistent with previous studies, we found that a history of mental illness intensified current mental health conditions [1, 11]. There were severe inequalities in healthcare access for patients with mental illness in Bangladesh [21, 51]. The movement restriction worsened this gap by severely disrupting access to basic facilities and hospitals outdoors for patients with mental illnesses. The Government hospitals and some private healthcare facilities only served the COVID-19 cases. Hence, people with existing mental health conditions could not access the medical support they used to get before the pandemic, which could deteriorate their existing symptoms.

Quite unexpectedly, the results showed that being infected personally did not affect our participants' mental health, but the infection of other family members did. Looking after an infected family member seemed more challenging during the pandemic. Further investigation is needed to demystify the family and personal significance of mental health in a collectivist society like Bangladesh [52]. It would be interesting to explore if the Bangladeshi people's mental health depends on the well-being of their family members. As a family, do they hold such a strong emotional bonding that overshadows the concern for themselves with that for the family members? As expected, a double dose vaccination is significantly associated with lower anxiety, depression, and stress.

As observed in other studies, our results showed a clear link between mindfulness and mental health during the pandemic [12, 53]. After adjusting the effect of all other factors, mindfulness appeared to be the strongest predictor of better mental health. This result is in line with the previous studies [13, 15]. When the uncertainty about COVID-19 prevails everywhere, being mindful of here and now became the single most effective self-help strategy for sustained psychological well-being. We also found modest support for optimism concerning better mental health.

### Limitations

This study had several limitations. First, we could not establish any cause-and-effect relationship because we conducted a cross-sectional study. Second, despite much care, the recruitment of samples online could be biased. While online data collection was the only option during the strict movement restriction, it nonetheless excluded the chance of participation for individuals who had no internet connection. Third, the Bangla PSS-4 and LOT-R scales appeared to have poor psychometrics for the current sample.

### Implications and future direction

The findings of this study have several practical implications. First, mental health interventions could prioritize women, unemployed and unmarried adults. The vaccination drive could be strengthened, especially in rural areas, as this would ease the burden of worse mental health. Mental health interventions also needed to incorporate and scale-up mindfulness training for the mass. This could be done through community-based health programs already existing in the country [54]. Additionally, academic and professional organizations could arrange mindfulness-based training programs for students or employees [55]. However, we recommend that researchers utilize prospective design to investigate direct associations between explanatory and outcome measures.

## Conclusion

Bangladeshi adults reported increased stress, anxiety, and depression after one year of the COVID-19 pandemic. When planning mental health interventions, practitioners should prioritise those who are unemployed, women, and living with pre-existing mental health conditions. In addition, special attention should be given to people living in rural areas. Increased mindfulness and optimism through intervention could reduce the adverse effect of the pandemic on adult mental health.

## Supporting information

**S1 File. Univariate regressions.**
(DOCX)

## Acknowledgments

The authors acknowledge the contribution of study participants and the critical feedback supplied by the anonymous reviewers.

## Author Contributions

**Conceptualization:** Azharul Islam, Papia Mahbuba.

**Data curation:** Azharul Islam, Tanvir Ahmed.

**Formal analysis:** Azharul Islam.

**Funding acquisition:** Azharul Islam.

**Investigation:** Papia Mahbuba, Tanvir Ahmed.

**Methodology:** Papia Mahbuba, Tanvir Ahmed.

**Project administration:** Azharul Islam, Papia Mahbuba, Tanvir Ahmed.

**Software:** Azharul Islam, Shamsul Haque.

**Supervision:** Azharul Islam.

**Validation:** Shamsul Haque.

**Visualization:** Azharul Islam.

**Writing – original draft:** Azharul Islam, Shamsul Haque.

**Writing – review & editing:** Azharul Islam, Shamsul Haque.

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
