## [Decision Letter · Decision Letter 0]

7 Jun 2022

PONE-D-22-07194Modifiable and nonmodifiable factors associated with anxiety, depression, and stress after one year of the COVID-19 pandemicPLOS ONE

Dear Dr. Islam,

Thank you for submitting your manuscript to PLOS ONE. After careful consideration, we feel that it has merit but does not fully meet PLOS ONE’s publication criteria as it currently stands. Therefore, we invite you to submit a revised version of the manuscript that addresses the points raised during the review process.

Please note that we have only been able to secure a single reviewer to assess your manuscript. We are issuing a decision on your manuscript at this point to prevent further delays in the evaluation of your manuscript. Please be aware that the editor who handles your revised manuscript might find it necessary to invite additional reviewers to assess this work once the revised manuscript is submitted. However, we will aim to proceed on the basis of this single review if possible.  Your manuscript has been assessed by an expert reviewer, whose comments are appended to this letter. The reviewer has raised a number of concerns, in particular regarding the study design and statistical analysis, which must be addressed before the manuscript can be accepted for publication. In addition, we strongly recommend you have your manuscript copyedited by a colleague or a professional scientific editing service. 

We look forward to receiving your revised manuscript.

Kind regards,

Joseph Donlan

Editorial Office

PLOS ONE

Journal Requirements:

2. Please provide additional details regarding participant consent. In the ethics statement in the Methods and online submission information, please ensure that you have specified what type you obtained (for instance, written or verbal, and if verbal, how it was documented and witnessed). If your study included minors, state whether you obtained consent from parents or guardians. If the need for consent was waived by the ethics committee, please include this information

Reviewers' comments:

Reviewer's Responses to Questions

**Comments to the Author**

1. Is the manuscript technically sound, and do the data support the conclusions?

Reviewer #1: Yes

2. Has the statistical analysis been performed appropriately and rigorously? 

Reviewer #1: Yes

3. Have the authors made all data underlying the findings in their manuscript fully available?

Reviewer #1: Yes

4. Is the manuscript presented in an intelligible fashion and written in standard English?

Reviewer #1: No

5. Review Comments to the Author

Reviewer #1: The authors provide an interesting report on a study interested in modifiable and non modifiable factors associated with stress, anxiety and depression during the pandemic. The study was conducted in Bangladesh in May and June 2021, at a time most restriction were lifted, although new lockdowns occured in some regions starting June 22.

Introduction:

The introduction makes a good job at examining the prevalence of mental health issues during the pandemic. But it could be expected that the presented literature includes references related to depression and anxiety during past similar challenges (epidemic/pandemic). Framed as it is, the study doesn't present an ambition towards generalization to future events.

Lines 65 - 74: The authors report on 3 studies that were conducted in Bangladesh in relation to stress depression and anxiety during the pandemic in 2020. Were other studies conducted since ?

The authors should clearly define what they consider modifiable and non modifiable factors associated with mental health (they start doing so line 125) and frame their literature review from this perspective as this is an important part of their study. From reading the manuscript from beginning to end, I understand that mindfulness, optimism and pessimism are considered modifiable factors, whereas all other variables are not. This should be made clearer from the beginning.

It would be helpful for the readers that the authors included hypotheses at the end of the introduction, rather than the general aim to investigate "modifiable and non-modifiable factors" associated with mental health.

Methods:

Lines 203: The authors report the use of an unreliable instrument. The pessimism scale, with an alpha of .36 should be removed from the analyses. The optimism scale, with an alpha of .59, has a reliablity close to acceptable. I suggest it remains used unless another review considers this a more important flaw.

Lines 245 to 247. The authors use a median score to determine whether participants suffered from stress. From this operationalization, the prevalence of stress is by definition close to 50%.

The authors should indicate which statistical analyses were performed for each variable and justify their choice. Lines 248 ++, they mention the use of regressions, yet t tests are presented at the beginning of the results section.

If this is possible, a useful variable to include would be whether participants were living in a region that was in lockdown at the time of their participation.

Results:

Line 279. The variable "mental health status" has not been described before its use.

Lines 280 ++: The authors report on different statistical tests that should be presented in light of the authors research questions or relate to the sample description. The authors could start by mentioning something in the lines of "To examine modifiable and nonmodifiable factors associated with the considered mental health outcomes, we tested the association of XXXX with YYYY. For the tests related to the description of the sample, this should be better presented in the methods section - preferably in a table, examples of these are history of mental disorder, occupation, vaccination status, recovery from COVID etc. In any case the description of the sample (with or without tests) should be reported in the methods section. For instance see Table 1 in Varma, P., Junge, M., Meaklim, H., & Jackson, M. L. (2021). Younger people are more vulnerable to stress, anxiety and depression during COVID-19 pandemic: A global cross-sectional survey. Progress in Neuro-Psychopharmacology and Biological Psychiatry, 109, 110236. https://www.sciencedirect.com/science/article/pii/S0278584620305522?

Table 1 (in the authors manuscript): What are the tests performed (please present these in the methods section, with justification) ? The coefficients / test statistics should be presented in the table, not only the p-values.

Line 345++: Again, the prevalence of high stress directly derives from the authors' use of a median split. This is not informative.

The structure of the results section could be made more familiar by using a more frequently used organization in clinical psychology.

Discussion:

The authors could start by reminding the readers of their research questions. My opinion is the authors could try to relate their study to similarly stressful situation as the current pandemic (past research + try to provide implication for future situations), thereby making it more relevant for researchers interested in future stressors.

The authors should revise the language to improve readability. Only some examples are indicated below. I suggest the authors use proofreading services.

Lines 75, 76: Repharse "panicking too many people" (makes it seem there is an appropriate amount)

Lines 81 - 82: Rephrase "not equally lethal to all; some still maintained a healthy state" (lethal means deadly)

Lines 154-155: Rephrase "19 vital questions in the questionnaire" (vital means required to live)

Through out the manuscript: please use "mental disorder" instead of "mental illness"

Line 433: Rephrase 'mimic' (suggests the prevalence rates are capable of perceiving rates in earlier studies and to adapt to those)

6. PLOS authors have the option to publish the peer review history of their article (what does this mean?). If published, this will include your full peer review and any attached files.

Reviewer #1: No

---

## [Author Response · Author response to Decision Letter 0]

30 Aug 2022

We have prepared a separate document outlining our responses to each of the comment made by the reviewers.

---

## [Decision Letter · Decision Letter 1]

7 Feb 2023

PONE-D-22-07194R1Modifiable and nonmodifiable factors associated with anxiety, depression, and stress after one year of the COVID-19 pandemicPLOS ONE

Dear Dr. Islam,

Thank you for submitting your manuscript to PLOS ONE. After careful consideration, we feel that it has merit but does not fully meet PLOS ONE’s publication criteria as it currently stands. Therefore, we invite you to submit a revised version of the manuscript that addresses the points raised during the review process.

We look forward to receiving your revised manuscript.

Kind regards,

Eric Mayor

Guest Editor

PLOS ONE

Journal Requirements:

Additional Editor Comments (if provided):

Dear Authors,

I think you for your work in addressing the suggestions from the first round of reviews. I have now received the reviews from two reviewers for the revision. I think the paper is better without the reduction in word count suggested by Reviewer 1. I would suggest you do not limit the introduction to 700 words (this is your decision). Reviewer 2 has made some valid points with regards to the inclusion of recent papers.

Best wishes,

Eric Mayor

Reviewers' comments:

Reviewer's Responses to Questions

**Comments to the Author**

1. If the authors have adequately addressed your comments raised in a previous round of review and you feel that this manuscript is now acceptable for publication, you may indicate that here to bypass the “Comments to the Author” section, enter your conflict of interest statement in the “Confidential to Editor” section, and submit your "Accept" recommendation.

Reviewer #2: (No Response)

Reviewer #3: (No Response)

2. Is the manuscript technically sound, and do the data support the conclusions?

Reviewer #2: Yes

Reviewer #3: Yes

3. Has the statistical analysis been performed appropriately and rigorously? 

Reviewer #2: Yes

Reviewer #3: Yes

4. Have the authors made all data underlying the findings in their manuscript fully available?

Reviewer #2: Yes

Reviewer #3: Yes

5. Is the manuscript presented in an intelligible fashion and written in standard English?

Reviewer #2: Yes

Reviewer #3: Yes

6. Review Comments to the Author

Reviewer #2: Thank you for invitation to review Modifiable and nonmodifiable factors associated with anxiety, depression, and stress after one year of the COVID-19 pandemic. The manuscript was clear and methodologies intact and add value to the literature. Here minor changes requested to improve.

Introduction:

To long (1190) need to reduce it to be with the 700 words only.

Method

Sample description:

This section should be at result section

General Comments:

All other section was clear and straight forward.

Reviewer #3: The authors of the article conducted a study to investigate the predictors of anxiety, depression, and stress among Bangladeshi adults one year into the COVID-19 pandemic. The study recruited 1897 participants and used various self-reported measures to assess mental health outcomes, mindfulness, and optimism. My appreciation for successful completion of the study.

The changes in the articles after the first reviewer’s comments, as shown in the Revision 1 manuscript, solve a majority of the issues in the article.

My comment would be, to include more recent studies in the discussion section. A tabulated or textual representation and comparison of all recent studies exploring the mental health implications and its predictors would add more meaning to the discussion section. Articles after 2021 have not been highlighted. I understand that this article restricts itself to 1 year post-pandemic but, it would be informative to explore the post covid angle in more recent publications as well.

Also, a comparison of the situation of High-income countries and Low and middle-income countries would be interesting. At least a few sentences with meaningful references would be good.

7. PLOS authors have the option to publish the peer review history of their article (what does this mean?). If published, this will include your full peer review and any attached files.

Reviewer #2: No

Reviewer #3: **Yes: **Dr. Adrija Roy

---

## [Author Response · Author response to Decision Letter 1]

7 Mar 2023

We supplied a separate response to reviewers document.

---

## [Editor Report · Decision Letter 2]

9 Mar 2023

Modifiable and nonmodifiable factors associated with anxiety, depression, and stress after one year of the COVID-19 pandemic

PONE-D-22-07194R2

Dear Dr. Islam,

We’re pleased to inform you that your manuscript has been judged scientifically suitable for publication and will be formally accepted for publication once it meets all outstanding technical requirements.

Kind regards,

Eric Mayor

Guest Editor

PLOS ONE

Additional Editor Comments (optional):

Dear Authors,

I thank you for your resubmission of the manuscript. I find this version can be published as is and therefore have decided not to send it to reviewers prior to making my decision. I congratulate you on the acceptance of your manuscript.

With best wishes,

Eric Mayor
---

## [Editor Report · Acceptance letter]

15 Mar 2023

PONE-D-22-07194R2 

Modifiable and nonmodifiable factors associated with anxiety, depression, and stress after one year of the COVID-19 pandemic 

Dear Dr. Islam:

I'm pleased to inform you that your manuscript has been deemed suitable for publication in PLOS ONE. Congratulations! Your manuscript is now with our production department. 

Kind regards, 

on behalf of

Dr. Eric Mayor 

Guest Editor

PLOS ONE